# Lactucin, a Bitter Sesquiterpene from *Cichorium intybus*, Inhibits Cancer Cell Proliferation by Downregulating the MAPK and Central Carbon Metabolism Pathway

**DOI:** 10.3390/molecules27217358

**Published:** 2022-10-29

**Authors:** Khandaker Md Sharif Uddin Imam, Yu Tian, Fengjiao Xin, Yingying Xie, Boting Wen

**Affiliations:** 1Laboratory of Biomanufacturing and Food Engineering, Institute of Food Science and Technology, Chinese Academy of Agricultural Sciences, Beijing 100193, China; 2Institute of Medicinal Plant Development, Chinese Academy of Medical Sciences & Peking Union Medical College, Beijing 100193, China

**Keywords:** lung cancer, adenocarcinoma, NSCLC, apoptosis, cell cycle, ABPP

## Abstract

Lung cancer, especially adenocarcinoma, is the second most occurring and highest fatality-causing cancer worldwide. Many natural anticancer compounds, such as sesquiterpene lactones (SLs), show promising anticancer properties. Herein, we examined Lactucin, an SL from the plant *Cichorium intybus*, for its cytotoxicity, apoptotic-inducing, cell cycle inhibiting capacity, and associated protein expression. We also constructed a biotinylated Lactucin probe to isolate interacting proteins and identified them. We found that Lactucin stops the proliferation of A549 and H2347 lung adenocarcinoma cell lines while not affecting normal lung cell MRC5. It also significantly inhibits the cell cycle at G_0_/G_1_ stage and induces apoptosis. The western blot analysis shows that Lactucin downregulates the MAPK pathway, cyclin, and cyclin-dependent kinases, inhibiting DNA repair while upregulating p53, p21, Bax, PTEN, and downregulation of Bcl-2. An increased p53 in response to DNA damage upregulates p21, Bax, and PTEN. In an activity-based protein profiling (ABPP) analysis of A549 cell’s protein lysate using a biotinylated Lactucin probe, we found that Lactucin binds PGM, PKM, and LDHA PDH, four critical enzymes in central carbon metabolism in cancer cells, limiting cancer cells in its growth; thus, Lactucin inhibits cancer cell proliferation by downregulating the MAPK and the Central Carbon Metabolism pathway.

## 1. Introduction

Lung cancers are the second most occurring cancers and the primary cause of cancer-related death among both men and women worldwide [1]. In 2018, out of 9.6 million cancer-related deaths, 1.76 million were caused by lung cancers [2]. Non-Small Cell Lung Cancer (NSCLC) accounts for 85% of all lung cancers [3], and adenocarcinoma is the principal subtype makeup for 75–85% of lung cancer and related deaths [4]. 

Rapid metastasis of lung adenocarcinomata [5] are primarily responsible for their late diagnosis, significantly affecting patient survival [6]. Common causes of lung cancers include smoking, air pollution, occupational exposure, and genetics [6]. However, diet also plays a crucial chemoprophylactic role in lung cancers [7,8,9]. It is more evident from the increasing number of novel anticancer compounds discovered in common foods and drinks [10]. Besides rapid metastasis, the aggressive chemotherapeutic regimen often associated with late-stage lung cancer treatment also contributes to poor patient survival [11]. Natural products used as adjuvant therapy have promoted patient survival and quality of life [12,13]. These have created a demand for natural chemopreventive, chemoprotective, and adjuvant compounds in cancer treatment [3,12,13,14]. 

Natural products are an abundant source of novel therapeutics [14]. More than 67% of novel anticancer and antitumor drugs are natural compounds or their analog [14,15], and more than 200 compounds and derivatives are at different stages of drug development and trials [15]. Chicory (*Cichorium intybus* L.) is a widely distributed plant used as medicine, food, drink, and fodder [16]. It has been the subject of many pharmacological studies [16]. Recently, we reviewed the cytotoxic of Chicory [16], where 31 out of 87 *C. intybus* metabolites possess anticancer, antitumor, and related bioactivities. Lactucin is a phytometabolite from Chicory, phytochemically categorized as bitter Sesquiterpene Lactones (SLs). SLs have been a subject of interest in cancer research for decades, and many have reached clinical trials [15]. As indicated in Ghantous, Gali-Muhtasib, Vuorela, Saliba, and Darwiche [15] reviews, SLs bioactivity is strictly linked to three conserved structural features, e.g., (i) alkylating center reactivity, (ii) side chain and lipophilicity, and (iii) molecular geometry and electronic features. Since many of the structural features that lactucin shares with other SLs have shown antitumor bioactivity against NSCLC [15], Lactucin will likely possess these properties. 

Some plants of the Asteraceae family commonly synthesize Lactucin. It is one of the ingredients of lactucarium, a milky white liquid secreted by several lettuce species, e.g., *Lactuca serriola*, *L. saligna*, *L. viminea*, *L. glareosa*, *L. sativa*, [17,18,19], etc. In 1970 Chicory root water extract was reported as a light-sensitive, highly potent antimalarial compound. Later Bischoff et al. [20] attributed it to Lactucin, making Chicory the most famous source of Lactucin. Besides antimalarial properties, Lactucin also exerts or potentially exerts anti-inflammatory [21], sedatives [17], anti-adipogenic [22], and anthelmintics [23] effects. Zhang et al. [24] reported Lactucin induces apoptosis and sub-G_1_ cell cycle arrest in HL-60 (human leukemia cancer) cells. Ren et al. [25] reported similar effect on KB (human epidermoid carcinoma; IC_50_ = 75 μM), and Bel 7402 (human hepatocellular carcinoma; IC_50_ = 55 μM) cells. Apart from these reports, the anticancer effect of Lactucin, especially on lung cancer, remains largely unexplored. A structural activity study (SAR) by Ren, et al. [25] revealed the importance of an ester group (γ-butyrolactone) and one exocyclic methylene group for the antitumor activity of Lactucin-like Guanolides. Wang, et al. [22] reported that Lactucin inhibits adipogenesis by downregulating the JAK2/STAT3 signaling pathway and subsequent clonal expansion. However, the cellular target protein of Lactucin and the affected pathway in antitumor activity is still unknown. Herein, we evaluated the anticancer potency of Lactucin using A549 and H2347 lung adenocarcinoma cell lines in vitro to identify interacting proteins and underlying molecular mechanisms.

## 2. Results

### 2.1. Lactucin Inhibits Lung Adenocarcinoma Cells Proliferation

The effect of incrementing concentrations of Lactucin on normal lung cells and lung adenocarcinoma cells upon 24 h exposures was evaluated by 3-(4,5-dimethylthiazol-2-yl)-2,5-diphenyltetrazolium bromide (MTT) assay. Compared to respective control groups, higher Lactucin doses significantly inhibit A549 (IC_50_ = 79.87 μM) and H2347 (IC_50_ = 68.85 μM) lung adenocarcinoma cells’ proliferation in a dose-dependent manner while not affecting MRC-5 cells’ proliferation, Figure 1A. However, at different exposure periods (12, 24, and 28 h), Lactucin showed significant inhibitory activity on A549 cell proliferation at 24 h and 48 h exposures at higher doses only, Figure 1B. Given that Lactucin can significantly inhibit lung adenocarcinoma cell proliferation at 24 h, exposure at a higher amount without affecting normal lung cells, we used respective IC_50_ concentrations of Lactucin for treating A549 and H2347 cells in further cytological experiments. A549 and H2347 cells treated with incrementing doses of Lactucin for 24 h showed a significant dose-dependent increase in the expression of LC3-II (17 kDa) in H2347 cells; and LC3-I (19 kDa) in A549 and H2347 cells, Figure 1C,D.

### 2.2. Effects of Lactucin on Cell Cycle Progression in Lung Adenocarcinoma Cells

Since Lactucin inhibited lung cancer cell proliferation, its cell cycle inhibitory property and related markers were analyzed. Cytometric analysis of DNA Content of 24 h lactucin treated (with respective IC_50_ concentrations) lung adenocarcinoma cells A549 and H2347 were performed. It revealed that Lactucin inhibits the cell cycle at the G_0_/G_1_ phase, Figure 2A. It was further supported by the western blot (WB) analysis of Lactucin-treated (24 h) adenocarcinoma (A549 and H2347) cell’s protein, Figure 2B. Both cyclins (cyclin B1 and cyclin D1) and cyclin-dependent protein kinases (CDKs) (CDK2 and CDK4), which regulate the cell cycle through mutual interactions, were dose-dependently downregulated, Figure 2C,D. However, p21 and p53, the two CDKs inhibitors we tested, were dose-dependently upregulated. This result suggests Lactucin inhibits the cell cycle of lung adenocarcinoma cells at the G_0_/G_1_ phase by upregulating CDKs inhibitor p21 and p53, downregulating cyclins and CDKs expression.

### 2.3. Effects of Lactucin on Apoptosis in Lung Adenocarcinoma Cells

The apoptosis-inducing capacity of Lactucin was measured by flow cytometry analysis of treated A549 and H2347 cells. As shown in Figure 3A, after 24 h incubation with respective IC_50_ concentrations of Lactucin, A549 cell’s apoptosis increased from 1.93% to 13.42%, and H2347 cell’s apoptosis rose from 1.42% to 40.70%. We also observed significant induction of early apoptosis in A549 (from 3.86% to 18.35%) and H2347 (6.07% to 17.36%) cells, Figure 3A. Apoptosis-related indicators such as Bax, Bcl-2 expression; and Caspase-3, and PARP activation, were also compared through WB, Figure 3B. We observed that Lactucin induced a dose-dependent increase in the concentration of both peptides of cleaved Caspase-3, Figure 3C, cleavage of PARP, expression of Bax, and inhibited the expression of Bcl-2, Figure 3D. The upregulation of Bax, downregulation of Bcl-2, and cleavage induction of Caspase-3 and PARP indicate that Lactucin has apoptosis-inducing properties.

### 2.4. Lactucin Affects the MAPK Signaling Pathway by Downregulating MEK, ERK Phosphorylation

Effects of Lactucin on metabolic markers of lung cancer cells were observed through WB, Figure 4A. Protein lysate from 24 h 80 μM Lactucin-treated A549 cells showed a different level of expression of metabolic markers, Figure 4B. Lactucin dose-dependently increased the expression of PTEN, a tumor suppressor protein that negatively regulates the activation of Akt to p-Akt, Figure 4 [26]. Dose-dependent MEK 1/2 and ERK activation inhibition indicate that Lactucin inhibits A549 cell proliferation by downregulating the MAPK/ERK pathway. We used a biotinylated Lactucin probe to confirm this and isolated Lactucin interaction proteins from A549 cells.

### 2.5. Lactucin Inhibits A549 Cell Proliferation by Downregulating MAPK/ERK and Central Carbon Metabolism Pathways

Natural Lactucin was conjugated with Propargylamine through Michael’s addition reaction. Structure of resulting Lactucin-Propargylamine conjugate [(3aR,4S,9aS,9bR)-4-hydroxy-9-(hydroxymethyl)-6-methyl-3-((prop-2-yn-1-ylamino)methyl)-3,3a,4,5,9a,9b-hexahydroazuleno[4,5-b]furan-2,7-dione] was confirmed using ^1^H Nuclear Magnetic Resonance (NMR), Figure 5A. The yield of the probe was estimated at 50% using High Performance Liquid Chromatography (HPLC). Cytotoxicity of the Lactucin probe and Propargylamine showed that the Lactucin probe exerts similar antiproliferation activity on A549 cells as natural Lactucin does both, doses dependent (Figure 5B) and time-dependent (Figure 5C) manners. In contrast, Propargylamine does not exert any significant bioactivity.

During the Activity-Based Protein Profiling (ABPP) assay, after Co-Immunoprecipitation (Co-IP), isolated protein lysate resolved on the Sodium Dodecyl Sulfate-Polyacrylamide Gel Electrophoresis (SDS-PAGE) showed five distinct bands in the treatment column after coomassie blue staining (Figure 6B). After testing these five bands using Liquid Chromatography with tandem Mass Spectrometry (LC-MS/MS) and analyzing them using MASCOT against 2019 human proteins. Many probable matches were found. Functional annotation clustering of LC-MS/MS results with the metabolic markers (upregulated and downregulated) data using DAVID Bioinformatics resources showed Lactucin interacts with several key enzymes of the “Central carbon metabolism in cancer” pathway (Kyoto Encyclopedia of Genes and Genomes, KEGG). These interacting proteins were identified as L01 = Phosphoglycerate mutase 1/2/4 (PGM, 70 kDa), L02 = Pyruvate kinase M1/2 (PKM, 58 kDa), L03 = Pyruvate dehydrogenase E1 subunit alpha 1/2 (PDH, 43 kDa), L04 = Lactate dehydrogenase A (LDHA, 37 kDa), but L05 was not identified. At the beginning of this manuscript, we defined cancer and tumor as proliferation disorders. However, they often show increased glucose and glutamine consumption, lactate secretion rate, glycolysis rate, and modified use of metabolic enzyme isoforms [27]. KEGG “Central carbon metabolism in cancer” pathway (Appendix A and Figure 6C) shows the difference in carbon metabolism between normal and malignant cells. Lactucin binding to PGM and PKM lower pyruvate synthesis, which binds to LDHA to lower Lactucin synthesis. Lactucin also binds to PDH, limiting the availability of Acetyl CoA entering the Tricarboxylic Acid (TCA) cycle and ultimately reducing lactate production, a hallmark of a metastasized tumor. Reducing PGM, PKM, and LDHA overexpression is attributed to the upregulation of p53 in response to DNA damage. From the KEGG pathway for NSCLC (Appendix A and Figure 6C), we see several oncogenes, such as p53, p21, and Bax, were upregulated; Cyclin D1, Akt, MEK, ERK, and CDK4 were downregulated.

## 3. Discussion

In the present experiment, we observed that Lactucin has significant antiproliferative, G_0_/G_1_ cell cycle arrest, and apoptosis-inducing properties on A549 and H2347 lung adenocarcinoma, not on MRC-5 normal lung cells. Firstly, Lactucin-induced time and dose-dependent inhibition of A549 and H2347 cells were observed, and then their IC_50_ values were determined to be 79.87 μM and 68.85 μM, respectively. According to WHO [28], for L-6 (rat skeletal myoblast) cells, compounds with an IC_50_ above 90 μM are not cytotoxic, IC_50_ between 2–80 μM is moderately cytotoxic, and IC_50_ below 2 μM is cytotoxic. For natural compounds, Ren, et al. [25] described an IC_50_ below 100 μM to be cytotoxic and reported Lactucin inhibits KB and Bel 7402 cells with an IC_50_ value of 75 μM and 55 μM, respectively. Lima, et al. [29] considered IC_50_ of <40 µg/mL and <4 µg/mL for plant extract and pure compounds respectively to be cytotoxic. According to American National Cancer Institute (NCI), except for fibroblast, an IC_50_ of ≤30 µg/mL of plant extract is cytotoxic, but they did not specify pure natural compounds. IC_50_ values we observed in adenocarcinomas can thus be considered cytotoxic. The LC3-II and LC3-II/LC3-I ratio increase is a classic sign of autophagy [30]. However, this is not always the case due to the lower sensitivity of LC3-I than LC3-II in WB and the higher degradation rate of LC3-II in the presence of lysosomal protease inhibitor [30,31]. We observed a dose-dependent increase of LC3-II in H2347 and LC3-I in A549 and H2347 cells. It was probably due to the concomitant increase in LC3 production and LC3I to LC3II conversion, rapid degradation of LC3-II, or lower detection of LC3-I in WB. In conclusion, LC3 WB results didn’t sufficiently indicate that Lactucin directly induces autophagy. LC3-I and LC3-II increase was probably due to autophagosome increase by some other means [30].

We found Lactucin induces G_0_/G_1_ cell cycle arrests from cell cycle analysis. Specific cell cycle-related proteins, such as cyclins and CDKs, positively upregulate the cell cycle, whereas CDKs inhibitors stop the Cyclins and CDKs unit assembly [32]. In this experiment, we observed the downregulation of Cyclin B1, Cyclin D1, CDK2, and CDK4, and the upregulation of p21 and p53. Downregulation of Cyclin D1 and upregulation of p53 were associated with G_0_/G_1_ cell cycle arrest and apoptosis [32], which coincides with our findings. Lactucin-induced inhibition of cell proliferation was also reported by Zhang, et al. [24] in human HL-60 leukemia cancer cells and by Wang, et al. [22] in mouse 3T3-L1 fibroblast cells. Though in 3T3-L1 G_0_/G_1_ cell cycle arrest was reported, [22] in HL-60 cells, Sub-G_1_ cell cycle arrest was reported [24].

We also observed an increase in early and late apoptosis in A549 and H2347 cells induced by Lactucin. Lactucin induces apoptosis by upregulating the expression of mitochondrial apoptosis-related proteins, such as c-Caspase, c-PARP, and Bax, while downregulating the expression of Bcl-2. Zhang, et al. [24] reported Lactucin induces apoptosis in HL-60 cells by swelling up mitochondria and endoplasmic reticulum (ER) on the transition electron microscope (TEM). Jang, et al. [33] reported that Lactucin induces ROS-mediated apoptosis in human renal cancer cell Caki-1 by downregulating Bcl-2 expression and CFLARL stability where Bcl-2 downregulation was at a transcriptional level caused by inactivation of the NF-κB pathway. However, we observed concurrent upregulation of PTEN and downregulation of Akt, MEK, and ERK phosphorylation narrowed down the Lactucin-influenced NSCLC pathway to PI3K/Akt and MAPK/ERK. McCubrey, et al. [34] suggested that ERK can activate the NF-κB transcription factor (nuclear factor immunoglobulin κ chain enhancer-B cell) by phosphorylating and activating inhibitor κB kinase (IKK) through an indirect mechanism. We also performed an ABPP assay on A549 protein lysate using a Lactucin-Propargylamine probe. Structural activity relationship (SAR) studies by Ren, et al. [25] revealed that in SL, the position 8 ester group (γ-butyrolactone) and the methylene group at exocyclic position 11 (α), play a significant role in antitumor activities of Lactucin-like guaianolides (Figure 5A) [16]. α methylene γ lactone, the’ enone’ or unsaturated carbonyl system (O=C–C=CH_2_) was also reported to increase the toxicity towards tumor cells [15]. While synthesizing the Lactucin-Propargylamine probe, we avoided the 8-ester group and 11 methylene group positions mentioned earlier. However, adding the alkynyl group to 11 (α), exocyclic methylene didn’t hinder its cytotoxicity against A549 cells (Figure 5B,C). It also suggests that Lactucin may simultaneously exert its activity through different groups. Proteins isolated using this probe were analyzed in LC-MS/MS and combined with WB results for functional enrichment. In the DAVID functional enrichment of western blot and ABPP peptides, four were identified as key enzymes in central carbon metabolism in the cancer cell (Figure 6C). Overexpressed and modified central carbon metabolism enzymes are essential to support the exponential growth of the metastasized tumor. Regulating their expression will limit the available energy and thus control or seize tumor growth. 

In anticancer therapy, oncogenes and tumor suppressors are two likely targets for inhibiting cancer cells [26]. From the KEGG pathway of NSCLEC by Kanehisa Laboratories, we know that in the case of NSCLC like A549 or H2347, we know the potential oncogene and tumor suppressors targets. DAVID enrichment results showed that in the NSCLC, Lactucin downregulates MEK-ERK pathway, resulting in the downregulation of CyclinD1, CDKs, and upregulation of the p53. p53, in turn, upregulates p21 and Bax. It doesn’t show interaction with any of the upstream components of NSCLC. When DAVID enrichment for the KEGG pathway “Central Carbon Metabolism in Cancer” was plotted, it showed four Lactucin interacting enzymes besides the western blot findings. Downregulation of MAPK/ERK observed in Figure 6C also lowers the expression of c-Myc, a proto-oncogene involved in cell cycle progression, apoptosis, and cellular transformation. Low c-Myc inhibits uncontrolled DNA synthesis associated with cancer and thus lowers the expression of PGM, PKM, and LDHA, and eventually downregulates carbon consumption needed to sustain malignancy. Lactucin also binds to PDH, reducing its concentration, and decreasing Pyruvate to Acetyl CoA conversion, thus downregulating the TCA cycle and lactate production. Thus, Lactucin inhibits cell proliferation and simultaneously increases cell cycle arrest in lung adenocarcinoma by reducing the MAPK/ERK and lowering carbon metabolism-related enzyme availability.

## 4. Materials and Methods

Firstly, we tested the Lactucin for its cytotoxicity, cell cycle inhibiting, and apoptosis-inducing properties on lung adenocarcinoma cell lines. Then, its effect on cell cycle, apoptosis, and other related protein expression, was evaluated by WB analysis. Finally, we constructed a biotinylated Lactucin probe to extract Lactucin-interacting proteins, identified them using LC-MS/MS, and elucidated the molecular mechanisms.

### 4.1. Cell Materials

In the present study, we observed the effect of Lactucin (Shanghai Yuanye Biological Technology Co., Ltd., Shanghai, China) on two lung adenocarcinoma cell lines, namely, A549 (Cell Bank of Shanghai Institutes for Biological Sciences, Chinese Academy of Sciences, Shanghai, China), and H2347 (Chinese Academy of Medical Sciences and Peking Union Medical College, Beijing, China), as well as one normal lung cell line MRC-5 (National Infrastructure of Cell Line Resource, Beijing, China).

### 4.2. Cell Culture 

Frozen (liquid-vapor) cell lines were cultured in complete growth media [A549 and H2347 cells in Roswell Park Memorial Institute 1640 (RPMI-1640) with 10% Fetal Bovine Serum (FBS) (Biological Industries, Kibbutz Beit Haemek, Israel) and 1% Penicillin/Streptomycin (Caisson Laboratories, Smithfield, VA, USA); MRC-5 in Minimum Essential Medium (MEM HyClone, Chicago, IL, USA), with 10% FBS, 1% Penicillin/Streptomycin, and 1% Nonessential Amino acids (10 mM 100×, Solarbio Biotechnology, Beijing, China) (EMEM)] [26]. For freezing A549, H2347, and MRC-5 cell line, 10% Dimethyl sulfoxide (DMSO, MP Biomedicals, Shanghai, China) in FBS, RPMI-1640, and MEM complete growth medium, respectively, was used. Adherent cells were detached using 1–3 mL of 0.25% Trypsin EDTA (Gibco, Thermo Fisher Scientific, Waltham, MA, USA) for 2–4 min at room temperature (RT).

### 4.3. Cell Viability Assay

Subconfluent cells (<90%) were aseptically trypsinized and diluted to 5 × 10^4^ cell/mL using complete growth media RPMI-1640 (for A549 and H2347) or EMEM (for MRC-5). Cells were then seeded aseptically in 96 welled plates at 1 × 10^4^ cells/well (200 µL/well) and incubated for 12 h at 37 °C in a 5% CO_2_ incubator before dosing. After 12 h, we exposed the cells to different concentrations of Lactucin for 24 h. All Lactucin concentrations were prepared using complete growth media and contained equal vehicle volume DMSO (*v*/*v*). Cell viability was determined using a MTT cell proliferation kit (M1020, Solarbio Biotechnology, Beijing, China) following the manufacturer’s instructions. UV absorbance was scanned at 490 nm using a Spark microtitration plate reader (Tecan, Männedorf, Switzerland). Each experiment was repeated three times, the viability of the control group was set to 100%, and cell viability and IC_50_ values were calculated.

### 4.4. DNA Content/Cell Cycle Assay

Subconfluent A549 and H2347 cells (<90%) were cultured at 1 × 10^5^ cell/mL concentration using complete growth media RPMI-1640 for 12 h at 37 °C in 5% CO_2_. Cells were then treated with respective IC_50_ Lactucin solution or equivalent volume DMSO (*v*/*v*) solution and incubated for 24 h. After treatment, cells were trypsinized, washed, and fixed for 24 h. Fixed cells were washed and stained using a Propidium Iodide (PI) Flow Cytometry kit (Abcam, Cambridge, UK) following manufacturer instructions. The DNA contents were scanned using a flow cytometer (CytoFLEX, Beckman Coulter Inc., Miami, FL, USA). 

### 4.5. Apoptosis Assay

Subconfluent A549 and H2347 cells (<90%) were cultured at 1 × 10^5^ cell/mL concentration using complete growth media RPMI-1640 for 12 h at 37 °C in 5% CO_2_. Cells were then treated with respective IC_50_ Lactucin solution or equivalent volume DMSO (*v*/*v*) solution and incubated for 24 h. After treatment, cells were trypsinized and washed. Following manufacturer instructions, cells were double-stained using the FITC Annexin V Apoptotic kit (BD Pharmingen, San Diego, CA, USA). Samples were filtered (70 μM Nylon cell strainer, Falcon, Corning, Durham, NC, USA) and scanned using a flow cytometer.

### 4.6. Protein Extraction

5 × 10^6^ cell/mL untreated [A549 for Co-IP] and treated (Lactucin and DMSO in A549 and H2347 for WB) cells were washed (1× PBS chilled), scraped (cell scraper, Corning, Durham, NC, USA), collected in 1.5 mL centrifuge tubes, and then washed again twice (1× PBS chilled). Between each washing, cells were centrifuged at 600× *g* for 5 min at 4 °C. After the final centrifuge, 300–500 µL of chilled modified RIPA buffer (1% PMSF + 1% SDS in RIPA) was added to each tube, briefly vortexed and then homogenized using probe sonicator at 30% power of 3 × 5 s with a 10 s gap in between. Homogenized cells were then left to chill on ice for 1 h. After 1 h, cells were centrifuged at 14000× *g* at 4 °C for 10 min, and intact cells and nuclear materials were separated by pipetting the supernatant into new centrifuge tubes and stored at −20 °C. The protein concentrations were determined using the Bicinchoninic Acid (BCA) protein assay kit (Solarbio Biotechnology, Beijing, China) following manufacturer instructions. UV absorbance was scanned at 562 nm using a microtitration plate reader.

### 4.7. Western Blot Assay

Western blot was performed using the method adopted from Lu. et al. [26]. A549 and H2347 cells protein (10 µg/lane) along with a pre-stained page ruler (5 µg/lane) were resolved using 8, 10, and 12% SDS-PAGE, along with pre-stained protein ladder (#26617; Thermo Fisher Scientific, Carlsbad, CA, USA) and transferred to polyvinylidene fluoride (PVDF) transfer membrane (Immobilon^®^-P, Billerica, MA, USA) by wet transfer at 350 mA for 70–110 min (depending on targeted proteins molecular weight). PVDF membranes were washed thrice using Tris-buffered saline with 0.1% Tween^®^ 20 Detergent (TBST) (10 min each) and blocked using 5% Bovine Serum Albumin (BSA) in TBST (10 mM Tris-HCl, 150 mM NaCl, 0.1% tween 20) at room temperature (RT) for 1 h. The PVDF was then washed thrice using TBST (10 min each), cut, and probed using primary antibodies at 4 °C overnight. The primary antibodies include anti-Akt (AA326), anti-Bax (AB026), anti-cleaved Caspase-3 (AC033), anti-CDK-4 (AC251-1), anti-Cyclin D1 (AC853-1), anti-p53 (AF0255), anti-ERK1/2 (AF1051), anti-MEK1/2 (AF1057), anti-cleaved-PARP-1 (AF1567), anti-mTOR (AF1648), anti-Ki67 (AF1738), anti-p-ERK (AF1891), anti-p-Akt1/2/3(Thr 308) (AF5734), anti-p21 (AP021-1), anti-PTEN (AP686) from Beyotime Biotechnology (Shanghai, China); anti-LC3B (NB100-2220) from Novus Biologicals (Littleton, CO, USA); anti-β-actin (SC47778), anti-Cdk-2 (SC6248), anti-Cyclin B1 (SC245), anti- p-MEK1/2 (SC81503), anti-Bcl-2 (SC7882) from Santa Cruz Biotechnology (Santa Cruz, CA, USA). Next, the PVDF was washed thrice using TBST (10 min each) and probed with appropriate goat anti-mouse (ab150113) or anti-rabbit (ab97051) secondary antibodies (Abcam, Cambridge, UK) for 1 h at RT. Finally, the PVDF was washed (thrice for 10 min each), exposed using luminol reagents (Millipore, Billerica, MA, USA), and photographed using the CLiNX Chemiluminescence imaging system (Shanghai, China). 

### 4.8. Synthesis of Lactucin Probe

To synthesize the biotinylated Lactucin probe, Lactucin was conjugated with an alkynyl group through Michael’s addition reaction, Figure 6A. To a solution of lactucin (5.0 mg, 0.018 mM) in dry pyridine (0.3 mL), Propargylamine (1.5 mg, 0.027 mM) (inno-chem, Beijing, China) was added and stirred at 0 °C for 36 h under N_2_ air. The reaction was monitored by thin-layer chromatography (TLC). The solvent was concentrated and purified through column chromatography (Dichloromethane-CH_3_OH, 15:1) and dried. For structural confirmation, the probe was analyzed using ^1^H-NMR at 600 MHz (Oxford NMR AS600, Abingdon, UK) in deuterated DMSO (d-DMSO, Merck KGaA, Darmstadt, Germany), Figure 5A. The yield of the probe was estimated using HPLC. 

### 4.9. Cytotoxicity Test and Protein Labelling by Lactucin Probe

Cytotoxicity of the Lactucin probe and Propargylamine was examined by incubating A549 cells with incriminating (0 to 150 μM) concentrations and incubation periods (12, 24, and 48 h) of each. The total A549 protein lysate was prepared using the method described in Section 4.6. The Lactucin binding proteins were labeled and pulled out using a biotinylated Lactucin probe following the ABPP method adopted from the method described by Speers and Cravatt [35]. Two 400 µL of protein lysate (2 mg/mL in PBS) were aliquoted into a 1.5mL microcentrifuge tube, and Lactucin-probe or Propargylamine was added (final concentration 100 μM). Both tubes were incubated at RT overnight in a shaker. It was followed by the addition of biotin-azide (Institute of Medicinal Plant Development, Beijing, China) (final concentration 100 μM), then Tris (2-carboxyethyl) phosphine (TCEP, T1656) (final concentration 1 mM), and Tris [(1-benzyl-1H-1,2,3-triazole-4-yl)methyl]-amine (TBTA, T2993) (final concentration 100 μM) (Tokyo Chemical Industry, Japan), and Copper sulfate pentahydrate (CuSO_4_·5H_2_O) (final concentration 1 mM) (Merck KGaA, Darmstadt, Germany) to each tube with vortex after each addition. Both tubes were incubated at room temperature for 1 h with a vortex every 30 min. Then the tubes were centrifuged at 12,000× *g* for 10 min at 4 °C, and the supernatant was removed. Protein precipitate was dissolved by adding 750 µL of pre-cooled methanol and sonicating for 3–4 s at 4 °C using a probe sonicator (~30% power level). It was followed by methanol wash thrice with centrifugation (12,000× *g*, 4 °C for 10 min) in between.

### 4.10. Streptavidin Enrichment

Streptavidin enrichment was by method adopted from the method described by Speers and Cravatt [35]. Probed protein precipitates were dissolved in PBS using 600 µL of 0.2% SDS. Then 100 µL of streptavidin agarose resin (Pierce™ 20347, Thermo Fisher Scientific, Waltham, MA, USA) was added and mixed for 1 h in a shaker at RT. It is followed by washing with 1 mL of 1× PBS with centrifugation (for 1 min at 2500× *g*) in between and each time collection of supernatants. Finally, the streptavidin beads were eluted by adding 100 µL of 2× SDS-loading buffer, boiling in a water bath for 10 min. Cooling at 4 °C and centrifuging at 2500× *g* for 1 min. The supernatants were collected and stored at −20 °C until use.

### 4.11. SDS-PAGE and Staining

Protein samples collected from the previous step were resolved in 10% SDS-PAGE gel and visualized by boiling in 0.25% Coomassie Brilliant Blue (CBB) stain for 5 min [26]. The blue-stained gel was washed twice by boiling it with distilled water for 5–10 min. Protein bands from sample columns were cut and stored (at −20 °C) until the identification experiment.

### 4.12. LC-MS/MS Analysis

Resolved protein samples were analyzed using QTRAP^®^ 6500 LC-MS/MS System (AB Sciex LLC, Framingham, MA, USA). Mass spectrometric data were searched against the NCBI database with a taxonomy restriction to 2019 human proteins (172,061 sequences; 53,783,369 residues, 27 July 2020) using MASCOT V2.0 (Matrix Sciences, London, UK).

### 4.13. Data Analysis

A bar diagram was plotted for the cytotoxicity test to visualize the “mean percent of inhibition ± standard error” of cell proliferation using Microsoft^®^ Excel 2019 software. The significance of the lactucin doses was compared with control sample data and calculated by “one-way ANOVA” using “IBM^®^ SPSS^®^ Statistics 26”. Significant difference at *p* 0.05, 0.01, 0.001, and 0.0001 was calculated and expressed with different Asterix “*” numbers on the chart or the data table legend. IC_50_ was calculated by plotting the Log10 dose on *X*-axis versus the Normalized response on the *Y*-axis in Graph Pad Prism 7. Cell cycle cytometry data were analyzed using ModFit LT™ (Version 5.0.9). Apoptosis compensation calculation and data analysis were done using CytExpert™ (Version 2.3.0.84). Protein concentrations were calculated in Microsoft^®^ Excel 2019. WB bands were quantified using ImageJ^®^ (Version 1.52a) and analyzed using Graph Pad Prism 7. NMR data was analyzed, the structure was predicted using MestReNova™, and the chemical structure was made using ChemDraw^®^ Professional. LC-MS/MS results analyzed in MASCOT were combined with WB markers data, and functional enrichment was performed to predict the involved molecular pathway using DAVID Bioinformatics resources [36,37].

## 5. Conclusions

In developing anticancer therapy, discovering natural compounds and their mechanisms are the new frontier. They can concurrently affect multiple cell lines, and are helpful in attending to metastasized cancers, while still possessing molecular specificity towards different oncogenes and tumor suppressors. Herein, we reported cytotoxic, cell cycle inhibitory, and apoptosis-inducing properties of Lactucin in A549 and H2347 cells for the first time. We also narrowed down Lactucin’s potential target in the A549 cell as MAPK/ERK pathway. Lactucin exerts its effect by lowering the carbon metabolism necessary for metastasized tumor development. To the best of our knowledge, this study was the first attempt at identifying Lactucin’s anticancer mechanism in the lung cancer cell line. 

## Figures and Tables

**Figure 1 molecules-27-07358-f001:**
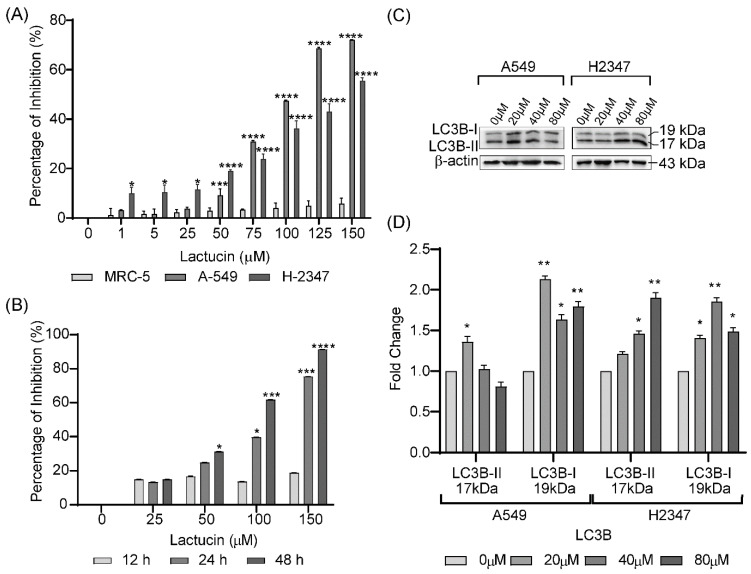
Lactucin significantly inhibits the proliferation of A549 and H2347 lung adenocarcinoma, while it doesn’t inhibit MRC-5 normal lung cell growth in a dose (**A**) and time (**B**) dependent manner. Lactucin significantly induces dose-dependent expression of LC3-II in H2347 cells (**C**,**D**), detected by WB analysis. Here * *p* < 0.05, ** *p* < 0.01, *** *p* < 0.001, and **** *p* < 0.0001 indicate significantly different from control.

**Figure 2 molecules-27-07358-f002:**
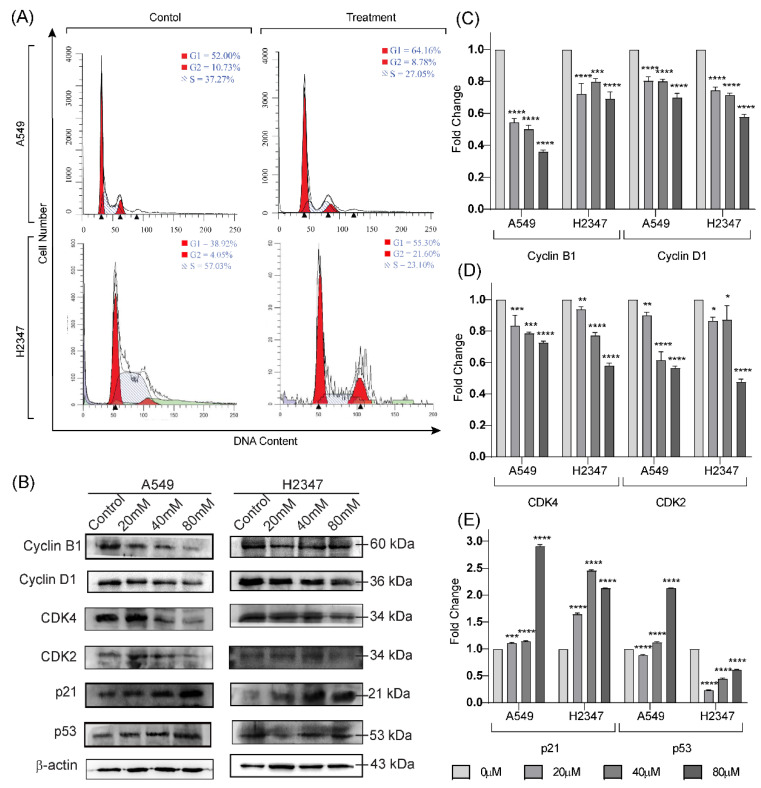
Lactucin and equivalent volume of vehicle (DMSO) exposure (24 h) induced cell cycle inhibition of lung adenocarcinoma A549 (80 μM), and H2347 (70 μM) cells (**A**) were measured by flow cytometry. WB analysis of cyclin B1, cyclin D1, CDK2, CDK4, p21, and p27 protein expression in A549 and H2347 cells after exposure to Lactucin at 0, 20, 40, and 80 μM for 24 h (**B**). Lactucin significantly downregulates Cyclin B1 and Cyclin D1 (**C**), CDK4, and CDK2 (**D**) and downregulates the p21 and p53 (**E**) expression in both A549 and H2347 cells in a dose-dependent manner. Data are expressed as the means ± standard deviation of triplicate experiments. * *p* < 0.05, ** *p*< 0.01, *** *p* < 0.001, and **** *p* < 0.0001 were considered significant.

**Figure 3 molecules-27-07358-f003:**
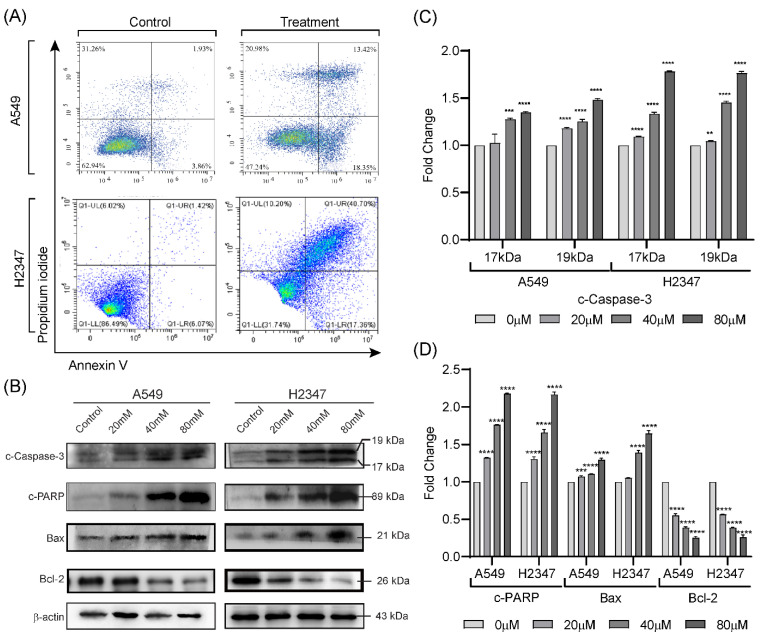
Effect of Lactucin on apoptosis induction of lung cancer cells. Lactucin and equivalent volume of the vehicle (DMSO) exposure (24 h) induced apoptosis of lung adenocarcinoma A549 (80 μM) and H2347 (70 μM) cells (**A**) were measured by flow cytometry. WB analysis of c-Caspase-3, c-PARP, Bax, and Bcl-2 protein expression was performed after A549 and H2347 cells were exposure to Lactucin at 0, 20, 40, and 80 μM for 24 h (**B**–**D**). Data are expressed as the means ± standard deviation of triplicate experiments. ** *p* < 0.01, *** *p* < 0.001, and **** *p* < 0.0001 indicate result’s significant difference from the control.

**Figure 4 molecules-27-07358-f004:**
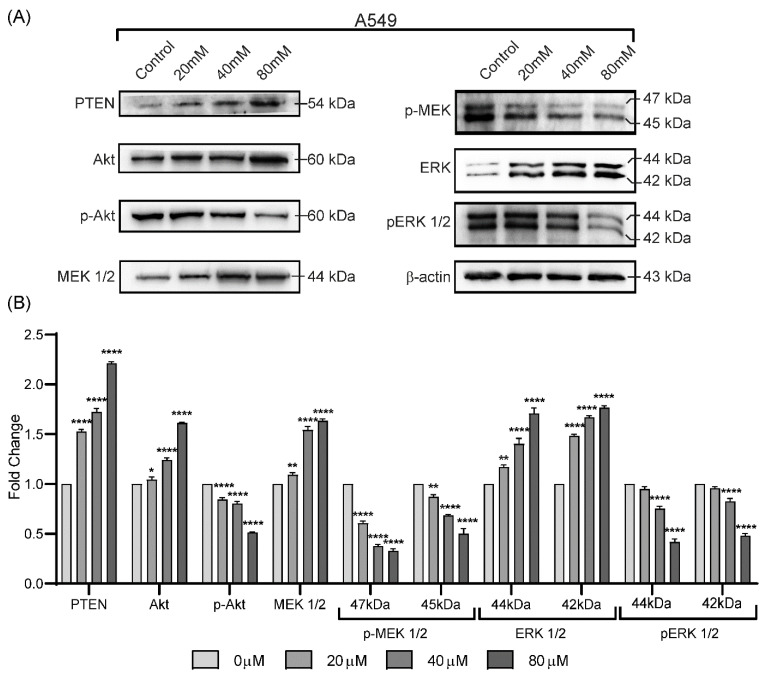
Lactucin inhibits A549 lung cancer cells by downregulating the MAPK/ERK pathway. (**A**) WB analysis of PTEN, Akt, p-Akt, MEK 1/2, p-MEK, ERK, and pERK 1/2 protein expression in A549 cell after exposure to Lactucin at 0, 20, 40, and 80 μM for 24 h. (**B**) Data are expressed as the means ± standard deviation of triplicate experiments. * *p* < 0.05, ** *p*< 0.01, and **** *p* < 0.0001 indicate result’s significant difference from the control.

**Figure 5 molecules-27-07358-f005:**
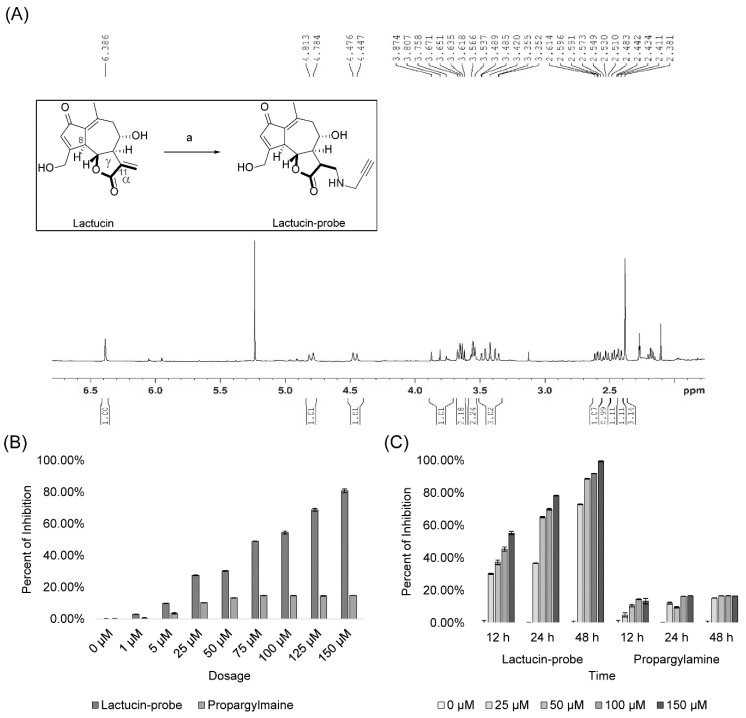
The structure of Lactucin-Propargylamine conjugate was confirmed using 1H NMR (**A**). Inset showing schematics of synthesis of Lactucin probe where reagents and conditions for synthesis were (a) propargylamine, pyridine, 0 °C, 36 h. Upon cytotoxicity evaluation on A549 cells, the Lactucin probe showed similar bioactivity as natural Lactucin compared to Propargylamine in incremental dose (**B**) and progressive dosing time (**C**) experiments.

**Figure 6 molecules-27-07358-f006:**
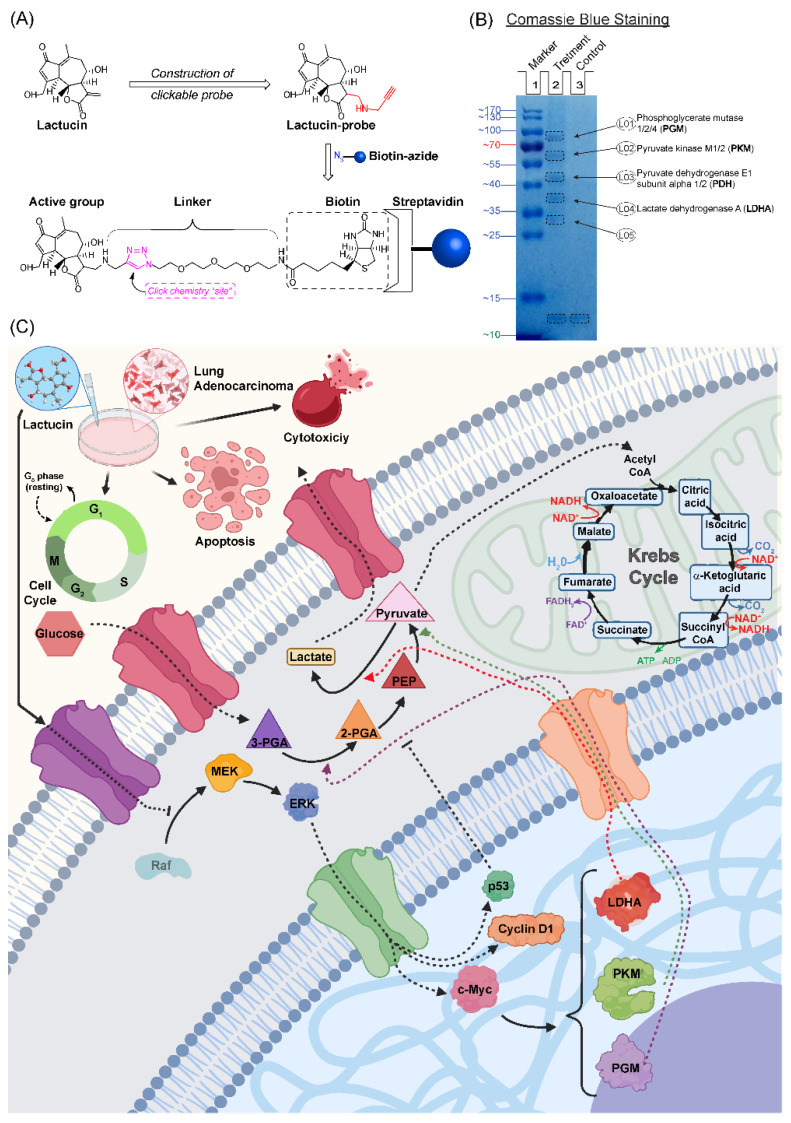
Lactucin downregulates Raf-MEK-ERK by inhibiting upstream trans-membrane peptides. Biotinylated Lactucin probes bound to targeted proteins are immobilized to streptavidin agarose through biotin, azide, and a linker molecule (**A**). Visualizing SDS resolved A549 protein lysate showed biotinylated Lactucin probe binds five peptides (**B**). Lactucin downregulates A549 lung adenocarcinoma cells Raf-MEK-ERK pathway and Central Carbon Metabolism in Cancer cells (**C**).

## Data Availability

The data presented in this study are available on request from the corresponding author.

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
