# Peer review of "Lactucin, a Bitter Sesquiterpene from Cichorium intybus, Inhibits Cancer Cell Proliferation by Downregulating the MAPK and Central Carbon Metabolism Pathway"

_molecules, 2022, doi:10.3390/molecules27217358_

Round 1

Reviewer 1 Report

The author studied the effects of Lactucin against lung cancer in vitro by determining cytotoxicity, apoptotic property, and cell cycle inhibitory capacity. The underlying mechanisms of anti-cancer effects of Lactucin were also discovered. Results showed that Lactucin exhibited anti-cancer effect by lowering the carbon metabolism necessary for metastasized tumor development. I have some concerns about this manuscript.

1. In abstract, the final word is "thus,", which looks like it is not an ending.

2. Does rutin have any effect on the growth of normal cells?  (instead of cancer cell), the author should offer some proofs.

3. Why did the author select the dose of 20 40 80 µM in the following research?

4. I think there is something wrong with the title of the graph, which can not fully include the content of verification. I suggest that the author can describe each graph directly, without an additional title.

5. Language need to be improved.

Author Response

Point 1: In abstract, the final word is "thus,", which looks like it is not an ending.

Response 1: Thank you for your comments. We apologize for the typo. The whole sentence is, "Thus; Lactucin inhibits cancer cell proliferation by downregulating the MAPK and the Central Carbon Metabolism pathway." which has been added in lines 34 – 35.

Point 2: Does rutin have any effect on the growth of normal cells? (instead of cancer cell), the author should offer some proofs.

Response 2: Thank you for your suggestion. In the present experiment, we studied Lactucin, not rutin.

On page 3, section 2.1, we showed that Lactucin dose and time-dependently inhibit adenocarcinoma cell A549 and H234 proliferation while not significantly affecting normal lung cell MRC5 (lines 94 – 97). We also included this statement in the abstract (page 1, lines 25-26).

Point 3: Why did the author select the dose of 20 40 80 µM in the following research?

Response 3: Thank you for your comment. From the cell viability assay (page 3, section 2.1, lines 94 – 97), we see that for Lactucin, the IC50 values of A549 and H2347 cells are 79.87 μM and 68.85 μM, respectively. Also, Figures 1A and B show that at lower doses (0 – 25 μM) and higher doses (125 – 150 μM), the adenocarcinoma cells population compared to control changes were less prominent. At lower doses, Lactucin concentration in the media was too low to affect the cells. The vehicle DMSO concentration was too high at higher concentrations to distinguish its effect from the control.

Figure 1B also shows that 12h exposure to Lactucin doesn't significantly differ between doses. In contrast, 48 h exposure slightly increases the inhibition. More prolonged exposure poses a greater risk of contamination.

So, we use 0, 20, 40, and 80 μM of Lactucin for 24 h in the cell cycle, apoptosis, and western blot experiments.  

Point 4: I think there is something wrong with the title of the graph, which can not fully include the content of verification. I suggest that the author can describe each graph directly, without an additional title.

Response 4: Thank you for your suggestion. We have modified all the figure's captions according to your suggestion. 

Point 5: Language need to be improved.

Response 5: Thank you for your suggestion. We checked the manuscript's language and made the necessary corrections.

Reviewer 2 Report

The authors of this manuscript consider a sesquiterpene lactone, lactucin, as a potential anticancer compound. The paper considers the cytotoxic effect of lactucin on adenocarcinoma cell lines A549 and H2347, as well as normal cells of lung fibroblasts. It is reported about the selective cytotoxic effect of lactucin against cancer cells, inhibition of the proliferation of these cells at the stage of inhibiting the cell cycle at the G0 / G1 stage, the action of this substance as an inducer of apoptosis, with the identification of the expression of the corresponding proteins. The work is undoubtedly of interest, but there are a number of recommendations for its improvement. Notes: 0. English needs improvement (for example, the article an is not put before a consonant, line 20) 1. There is no completeness in the abstract, the last sentence is not completed (line 32) 2. Please use italics when writing Latin names, e.g. “in vitro” (lines 84-85) 3. On page 3, which shows the WB result showing LC3B expression (Fig. 3C), it is desirable to indicate where LC3-I and LC3-II are, since the text refers to LC3-II, not LC3B (lines 99, 106 ) 4. The names of some molecules are misspelled, e.g. “c-Caspas-3” in Fig.3B, as well as “NF κB” (line 255) 5. There is no “control” in Fig.3C and 3D, and there is no mention of these figures in the caption under Figure.3 (line 144). Under Fig.4. also include a description for the block Fig.4A and Fig.4B 6. There is no information in the materials and methods about obtaining lactucin from Cichorium intybus, although the authors emphasize this in their work, judging by the title 7. The text of the manuscript contains obscure characters, such as “oC” (line 204) 8. Names of cell lines used must be without hyphens (A549 and H2347) 9. In the discussion, the authors write “dose-dependent increase of LC3-II in H2347 and LC3-I in A-549 and H-2347 cells” (lines 231,232), although the results say nothing about the change in LC3-I 10. There are gaps and errors in punctuation marks (line 93, line 464, etc.) 11. The ERK signaling pathway has several alternative abbreviations, the main ones being Ras-ERK, MAPK/ERK, Ras-Raf-MEK-ERK pathway. The authors use the abbreviation MEK-ERK, which is not generally recognized 12. Flow cytometry results show different frames for the same H2347 cells in control and experiment (Fig.3A). Thus, the manuscript contains a large number of flaws, many of which are not semantic. I recommend a major revision

Author Response

Point 1: English needs improvement (for example, the article an is not put before a consonant, line 20) 1. There is no completeness in the abstract, the last sentence is not completed (line 32).

Response 1: Thank you for your valuable comments. Using an indefinite article (a or an) before acronyms and abbreviations depend on how it sounds. Abbreviations and acronyms are sometimes spelled out as a word (e.g., FAO) or as an individual letter (e.g., BBC). In either case, if they sound like a vowel, an precedes it. Here, the acronym of the words “Sisquiterpine Lactone” is “SL” which is pronounced as “as al”; hence we used an before it.

We also checked the manuscript’s language and made the necessary corrections.

Point 2: Please use italics when writing Latin names, e.g. "in vitro" (lines 84-85).

Response 2: Thank you for your suggestion. We have italicized the Latin name in vitro (Page 3, Lines 88).

Point 3: Why did the author select the dose of 20 40 80 µM in the following research?

Response 3: Thank you for your comment. From the cell viability assay (page 3, section 2.1, lines 94 – 97), we see that for Lactucin, the IC50 values of A549 and H2347 cells are 79.87 μM and 68.85 μM, respectively. Also, Figures 1A and B show that at lower doses (0 – 25 μM) and higher doses (125 – 150 μM), the adenocarcinoma cells population compared to control changes were less prominent. At lower doses, Lactucin concentration in the media was too low to affect the cells. The vehicle DMSO concentration was too high at higher concentrations to distinguish its effect from the control.

Figure 1B also shows that 12h exposure to Lactucin doesn't significantly differ between doses. In contrast, 48 h exposure slightly increases the inhibition. More prolonged exposure poses a greater risk of contamination.

So, we use 0, 20, 40, and 80 μM of Lactucin for 24 h in the cell cycle, apoptosis, and western blot experiments.  

Point 4: On page 3, which shows the WB result showing LC3B expression (Fig. 3C), it is desirable to indicate where LC3-I and LC3-II are, since the text refers to LC3-II, not LC3B (lines 99, 106 )

Response 4: Thank you for your recommendation. We have made recommended changes in  Figures 1C and D.

Point 5: The names of some molecules are misspelled, e.g. "c-Caspas-3" in Fig.3B, as well as "NF κB" (line 255).

Response 5: Thank you for your comment. We apologize for the typo. "c-Caspase-3" in Figure 3B and "NF-κB" in line 275 has been corrected.

Point 6: There is no "control" in Fig.3C and 3D, and there is no mention of these figures in the caption under Figure.3 (line 144). Under Fig.4. also include a description for the block Fig.4A and Fig.4B.

Response 6: Thank you for the suggestion. In the Figure 3 caption, Figures C and D are added. Figures 3C and D show that the 0µM concentration of Lactucin (no treatment) serves as the control. In the Figure 4 caption, Figure 4A and B descriptions are added.

Point 7: There is no information in the materials and methods about obtaining lactucin from Cichorium intybus, although the authors emphasize this in their work, judging by the title.

Response 7: Thank you for your comment. In the "Materials and Method" section, Page 11. Lines 324 – 325, we have mentioned that the Lactucin we used was obtained from "Shanghai Yuanye Biological Technology Co. Ltd, Shanghai, China".

Point 8: The text of the manuscript contains obscure characters, such as "oC" (line 204).

Response 8: Thank you for your comments. We apologize for the typo. We have corrected the mentioned character to "°C", Line 222.

Point 9: Names of cell lines used must be without hyphens (A549 and H2347).

Response 9: Thank you for your valuable suggestion. We have corrected all the cell line names as recommended.

Point 10: In the discussion, the authors write "dose-dependent increase of LC3-II in H2347 and LC3-I in A-549 and H-2347 cells" (lines 231,232), although the results say nothing about the change in LC3-I.

Response 10: We appreciate your input. The findings of the LC3-I and LC3-II western blot of A549 and H2347 are shown in Figure 1D. Different Asterix "*" numbers are displayed on the bar graph to indicate the degree of significance of the deviation from the control. Here, in this instance *p < 0.05, **p < 0.01, ***p < 0.001, and ****p < 0.0001 indicate significantly different from control. This data led us to write, "We observed a dose-dependent increase of LC3-II in H2347 and LC3-I in A549 and H2347 cells."

Point 11: There are gaps and errors in punctuation marks (line 93, line 464, etc.).

Response 11: We appreciate your observation. Lines 98 and 485 of the document have the mentioned punctuation marks fixed. We are sorry for the error.

Point 12: The ERK signaling pathway has several alternative abbreviations, the main ones being Ras-ERK, MAPK/ERK, Ras-Raf-MEK-ERK pathway. The authors use the abbreviation MEK-ERK, which is not generally recognized.

Response 12: Thank you for your valuable suggestion. We have changed the abbreviation MEK-ERK to Raf-MEK-ERK and MAPK/ ERK.

Point 13: Flow cytometry results show different frames for the same H2347 cells in control and experiment (Fig.3A).

Response 13: Thank you for your comment. We have analyzed the flow cytometric data of the H2345 cell and put the same frame in control and treatment (Page 5, Figure 3A). As a result, the number of apoptotic and early apoptotic cells also changed slightly. We have also modified the respective description in the result section of the manuscript (Page 5, Line 144).

Reviewer 3 Report

The manuscript title “Lactucin, a bitter sesquiterpene from Cichorium intybus, inhibits cancer cell proliferation by downregulating the MAPK and Central Carbon Metabolism pathway” is well written and have systematic significance. The authors performed a detailed study on Lactucin (from Cichorium intybus) that significantly hinders cancer cell proliferation by downregulating the Carbon Metabolism (CM) and MAPK pathway.

My decision is accept after minor revision, specially, references must be added in the material and methods section. My minor suggestions for authors are as follows: 

Reviewer Comments:

1-      Line 21, 24, and so on: I suggest authors whenever you use any abbreviation than write it full name first time.

2-      The authors only used references in the 4.13 section of material and methods. From 4.1 to 4.12 sub-sections the authors didn’t use any reference why????? This is a big question, all the methods develop by the authors of this manuscript? Or they used previously published methods without references??? If is it, than please add references. If the authors used previous methods with some modifications, than please add reference first and then write …et. al., with slightly modifications etc……

Author Response

Point 1: Line 21, 24, and so on: I suggest authors whenever you use any abbreviation than write it full name first time..

Response 1: Thank you for your comments. We have replaced both acronyms with their respective pronunciation (Lines 21 and 25) and checked throughout the manuscript for abbreviation error and revised them.

Point 2: The authors only used references in the 4.13 section of material and methods. From 4.1 to 4.12 sub-sections the authors didn't use any reference why????? This is a big question, all the methods develop by the authors of this manuscript? Or they used previously published methods without references??? If is it, than please add references. If the authors used previous methods with some modifications, than please add reference first and then write …et. al., with slightly modifications etc……

Response 2: We appreciate your valuable comments. As recommended, we have revised the "Materials and Methods" section and made appropriate references except for some generic methodologies, such as western blot assay and LC-MS/MS analysis.

Round 2

Reviewer 1 Report

I have no comments at this time.

Author Response

Point 1: I have no comments at this time.

Answer 1: We appreciate your thoughtful advice. We also appreciate you taking the time to revise our manuscript.

Reviewer 2 Report

In the discussion, the authors write "dose-dependent increase of LC3-II in H2347 and LC3-I in A-549 and H-2347 cells" (lines 231,232), although the results say nothing about the change in LC3-I. This needs to be corrected in the text of the manuscript.

Author Response

Point 1: In the discussion, the authors write "dose-dependent increase of LC3-II in H2347 and LC3-I in A-549 and H-2347 cells" (lines 231,232), although the results say nothing about the change in LC3-I. This needs to be corrected in the text of the manuscript.

Response 1: We appreciate your input. We have explained this in response to your previous query (point 10).
The findings of the LC3-I and LC3-II western blot of A549 and H2347 are shown in Figures 1C and 1D. In figure 1D, different Asterix "*" numbers are displayed on the bar graph to indicate the degree of significance of the deviation from the control. Here, in this instance *p < 0.05, **p < 0.01, ***p < 0.001, and ****p < 0.0001 indicate significantly different from control.

On page 3, lines 103 -106 of the Results section, we mentioned the western blot experiment result of the LC3B. We mentioned, “A549 and H2347 cells treated with incrementing doses of Lactucin for 24 h showed a significant dose-dependent increase in the expression of LC3-II (17 kDa) in H2347 cells and LC3-I (19 kDa) in A549 and H2347 cell, a marker for autophagy, Figure 1C, D.”

On page 10, lines 245 – 253 of the discussion section, we briefly discussed the possible reasons for the results we found because, in the present experiment, we didn't find any direct evidence of autophagy from the LC3B activation. We mentioned, "We observed a dose-dependent increase of LC3-II in H2347 and LC3-I in A-549A549 and H-2347H2347 cells. It was probably due to the concomitant increase in LC3 pro-duction and LC3I to LC3II conversion, rapid degradation of LC3-II, or lower detection of LC3-I in WB."